# Performance Evaluation of Reconfigurable Intelligent Surface against Distributed Antenna System at the Cell Edge

**Nuraqila Alini Kamaruddin [1], Azwan Mahmud [1,*], Mohamad Yusoff Bin Alias [1], Azlan Abd Aziz [2] and Syamsuri Yaakob [3]**

1. Faculty of Engineering, Multimedia University, Cyberjaya 63100, Malaysia; 1171101726@student.mmu.edu.my (N.A.K.); yusoff@mmu.edu.my (M.Y.B.A.)
2. Faculty of Engineering & Technology, Multimedia University, Malacca 75450, Malaysia; azlan.abdaziz@mmu.edu.my
3. Faculty of Engineering, University Putra Malaysia (UPM), Serdang 43400, Malaysia; syamsuri@upm.edu.my
* Correspondence: azwan.mahmud@mmu.edu.my

**Abstract:** In small spaces in which typical cell towers cannot be constructed, distributed antenna systems (DASs) are the preferable approach for increasing network coverage, as they are a superior solution for congested, high-volume areas. However, due to their high cost and complexity in backhaul routing, reconfigurable intelligent surfaces (RISs) are a promising solution to overcome the major drawbacks of DAS systems while improving network coverage. Thus, this work investigated a correlation of execution in an RIS-aided system cell framework and DAS-aided system cell framework where a simple and precise structure for the performance measurement of area spectral efficiency (ASE) and energy efficiency (EE) under realistic channel presumptions was introduced. The analysis started with the downlink ergodic capacity with regards to the RIS framework and DAS framework under a generalized Nakagami-m fading channel with the presence of path-loss attenuation and interference from co-channel base stations (BSs), and was simplified further by utilizing a moment-generating function (MGF)-based approach. From the computed expression, the effects of traffic activity, EE and ASE were derived and analyzed for both systems in the presence of co-channel interference. The results were then verified by comparing them with Monte Carlo simulations, and the findings show that the two outcomes generally match. Based on these, it is demonstrated that the ASE performance of the RIS-assisted system in various traffic activity conditions outperforms the DAS-aided system; however, in high signal-to-noise (SNR) regions with full traffic activity, the ASEs are highest for both systems; by only 0.005 bits/s/Hz/km$^2$.

**Keywords:** area spectral efficiency (ASE); co-channel interference (CCI); distributed antenna system (DAS); energy efficiency; ergodic capacity; reconfigurable intelligent surface (RIS); traffic activity factor

## 1. Introduction

Fifth-generation (5G) wireless networks have already been installed in several countries as of mid-2019, and the first 5G mobile devices are already on the market. As the world is watching the 5G network start to take off, a new evolution of mobile communications has emerged, one that includes three use cases with distinct requirements: increased mobile Internet access, ultrareliable and low-latency communications, and vast machine-to-machine interactions. From this standpoint, the wireless industry is looking ahead to Beyond-5G (B5G), or sixth-generation (6G), technological breakthroughs in anticipation of new applications that includes the three use cases while fulfilling the increased demand in wireless communication and sensing networks with greater and greater agility, coverage, and throughput [1]. Previously, due to mobile data and cellular growth, it has become essential to install more small cells to counter the rapid increase in the number of mobile subscribers. This has opened the door for distributed antenna systems (DASs), a network of spatially separated antennas that are connected to common sources via transport medium

(coaxial line or fiber optic line) to provide wireless service within a geographic area or structure. Because DAS systems are substantially smaller than typical macro cell or cell tower counterparts, they are far superior solutions for congested, high-volume areas. DAS networks tend to give better network coverage than typical cell towers, in addition to taking up less space. The quality of a cellular signal degrades as the distance between a receiver and transmitter increases; however, to ensure that mobile users are always within close proximity to a strong and fast signal, DAS is used by strategically placing receivers in targeted locations.

However, there are some major drawbacks of utilizing the DAS system in the cellular communications network. When expanding the network coverage and bandwidth, one of the main factors for design planning is the cost. DAS systems, unfortunately, are not cheap. In fact, DAS networks are substantially more expensive than small cell networks [1]. The necessity for fiber-optic cables to connect each radio head to a central hub, as well as the creation of the hub and the location of the cellular base stations, all contribute to the price difference. Backhaul routing is also a concern for the DAS network. The installation process can be complicated because each of the radio heads used to broadcast the cellular signal requires routed fiber-optic connections. In order for the system to work, each of the radio heads must be connected to the central hub via fiber-optic cable. With all of these considerations in mind, researchers have started to uncover another breakthrough or alternative in enhancing the cellular coverage. In much the same way that DAS began as a promising research subject a few years ago and became a critical enabler for the evolution of commercial deployment from fourth-generation (4G) to 5G, reconfigurable intelligent surfaces (RISs) have attracted significant interest from academic and the telecommunication industry at this early stage of research [2].

RISs could be a promising option for creating a software-configurable smart radio environment. An RIS is a large, thin meta-surface made of metallic or dielectric material that is made up of a collection of passive subwavelength scattering elements with a particularly defined physical structure. Its elements can be manipulated in software to change the electromagnetic (EM) propagation properties such as phase shifting of the incident of reflected radio frequency (RF) signals. Other distinguished characteristics of RIS are as follows: RISs are made of cost-effective materials, are practically passive, and do not require a specific energy source [3–5]. They may also be simply installed on building facades, factory ceilings, and internal spaces because they have a physical appearance similar to that of a mirror. One of the most advantageous use cases for the RIS is in a restricted line-of-sight (LOS) path or at the cell edge, thanks to their ability to reconfigure signals [4]. The signal reflected by the RISs also non-invasively augments the environment of existing wireless networks for increased spectral efficiency and energy efficiency without the need to undergo the process of the wireless standards and design modifications for transceivers already in use [5].

RISs have been discovered as assisting sensing, localization, and wireless power transmission, in addition to improving the performance of wireless communications networks [2–4]. It could be especially useful for high-frequency band networks such as millimeter-wave and terahertz, where blockage and absorption-induced dead zones and channel rank deficiency are common performance bottlenecks. RIS-aided systems have recently been investigated with respect to energy efficiency [5], signal-to-noise ratio (SNR) maximization in a Rayleigh fading model [6], capacity maximization [7] and average bit-error rate (BER) in the presence of interference [8]. Further applications of RIS-enabled schemes in the Internet of Things (IoT) [9], cognitive radio networks [10] and massive multiple-input multiple-output (MIMO) [11] has been performed where RIS usage provides better performance in comparison to the non-RIS system. However, except for [5], where the authors specifically compared the RIS-assisted system with the relay-assisted system, other researchers only focused on the comparison between the RIS-aided system and non-aided system.

Given this motivation, there is a need to perform an analysis on the RIS-assisted cellular system and DAS-assisted cellular system, particularly for capacity maximation and coverage for hidden-node users at the cell edge. The objective of this paper is to perform an evaluation between the RIS-assisted cellular system in comparison to the DAS-assisted system, wherein we first present a new mathematical analysis for ergodic capacity of the downlink signal received by mobile users in the RIS- and DAS-aided cellular system in the Nakagami-m fading channel, with the presence of path loss and interference from the co-channel base station (BS). For the RIS system, a non-LOS path from home cell BS is considered where the received signals are combination of reflected paths via RIS. Presume RIS is capable of reflecting away the undesired signal from reaching the mobile user, the interfering component is the signal from co-channel BS to RIS. For the DAS system, mobile users received a combination of signals from collocated DASs in the home cell and interfering signals from DASs in the co-channel cell. Then, we simplify the analytical expression of ergodic capacity for both systems by utilizing the moment-generating function (MGF) in order to provide the expectation of the random variables (RV) and reduce the computation complexity in comparison to tedious and time-consuming simulation. We analyze the effect of the number of RIS elements, traffic activity factor and path loss to the ergodic capacity. We further perform evaluation on the performance of both systems, in terms of energy efficiency (EE) and area spectral efficiency (ASE). The main contributions of this paper are summarized as follows:

- Analysis of ergodic capacity and EE for the downlink signal received by mobile users in the RIS- and DAS-aided cellular system in the Nakagami-m fading channel, with the presence of path-loss attenuation and interference from the co-channel BSs.
- We simplify the expression of ergodic capacity for both system by utilizing the moment-generating function (MGF) in order to provide the expectation of the RV and reduce the integration number of the expressed formula.
- Application of RIS and DAS at the cell edge in a fractional frequency reuse cellular system, whereby the effect of the frequency reuse factor (FRF) for the cell edge is analyzed utilizing the ASE.

Monte Carlo simulations are provided to verify the accuracy of the analysis.

The paper is organized as follows: In Section 2, we describe the two system models under study, starting with RIS then moving onto DAS. In Section 3, the numerical and simulation results are presented. Finally, Section 4 concludes the paper.

## 2. System Model

### 2.1. Reconfigurable Intelligent Surface System Model

As a passive element that does not provide any amplification, RISs are practically placed close to either the BS or the user position. In this paper, without a loss of generality, RISs are arbitrarily distributed at the cell edge to assist the cell edge user as shown in Figure 1. We designed a RIS system that could provide maximum suppression of the co-channel interference; six RISs are considered to be arbitrarily distributed at the cell edge, which is closer to the cell-edge users [7]. R is the radius of the whole system while r is the distance between the reference BS ($BS_0$) and the reference RIS ($RIS_0$), which is fixed and equivalent to (2/3)R. The mobile device or reference user, $U$ is assumed to be randomly distributed inside the $RIS_0$ coverage area with a distance d and can only receive the interference reflected signal from the $RIS_0$ without having any LOS signal from the BS of the home cell. For the co-channel interference (CCI), as an intelligent surface, we assume that RISs do not forward the signal to users as all undesired signals are reflected away [6]. Therefore, we only consider the interfering signal from the $k_{th}$ co-channel (CC) BS to the $RIS_0$ reference user $U$ with a distance of $l_k$. Without a loss of generality, we also assume that RISs have ideal channel estimation and utilize a continuous phase shift.

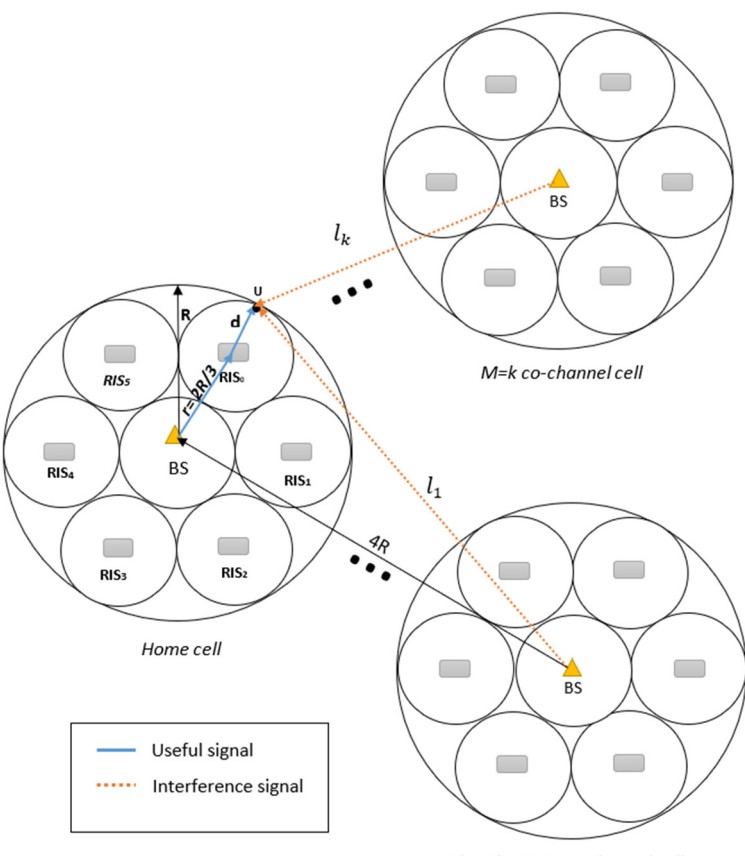

**Figure 1.** RIS-aided system model.

For the $RIS_0$ configuration with N number of reflective elements per RIS, the reflected signal received by $U$ in the presence of CCI from M neighboring cells can be expressed as [5,6]:

$$y_{RIS} = \sqrt{P}x_0 \left( \sum_{i=1}^{N} h_{s,i} \, h_{r,i} \, \sqrt{r^{-\beta_{s,i}} \, d^{-\beta_{r,i}}} \, e^{-j(\varnothing_i - \theta_{s,i} - \theta_{r,i})} \right) + \sum_{k=1}^{M} \sqrt{P}x_k h_{c,k} \sqrt{l_k^{-\beta_{c,k}}} \delta_k + w_0, \tag{1}$$

where the first term is the useful signal, the second term is the CC interference signals and $w_0$ represents the additive white Gaussian noise (AWGN) that satisfies $E\left[|w_0|^2\right] = \sigma^2$. $x_0$ and $x_k$ are the transmitted signals from useful and interfering signal from the $k_{th}$ BS, respectively, while $P$ is the transmitted signal power from any BS. $|h_{s,i}|^2$ and $|h_{r,i}|^2$ are the random channel gains from BS to $RIS_0$ and $RIS_0$ to $U$, respectively, while $|h_{c,k}|^2$ is the random channel gain from CC BS to $U$. $\theta_{s,i}$ and $\theta_{r,i}$ are the phase components of BS-$RIS_0$ and $RIS_0$-U while $\varnothing_i$ is the reconfigurable phase induced by the ith reflector of the RIS. $\beta_{s,i}$, $\beta_{r,i}$ and $\beta_{c,k}$ are the path loss exponents of BS-$RIS_0$, $RIS_0$-$U$ and CC-$U$, respectively, and $\delta_k$ ($k = 0, 1, \ldots, M$) is the binary RV that represents each channel of the interfering BS's traffic activity, where $\delta_k = 1$ if channel k is active and $\delta_k = 0$, otherwise.

Without a loss in generality, RIS is assumed to have the knowledge of the channel phase terms such that the RIS-induced phases can be adjusted to maximize the received SNR through appropriate phase cancellations and proper alignment of reflected signals from the intelligent surface, where $\varnothing_i = \theta_{s,i} + \theta_{r,i}$ for useful signals. Presuming that the values of all path-loss exponents in the equation are the same, where $\beta = \beta_{s,i} = \beta_{r,i} = \beta_{c,k}$, the

instantaneous signal-to-interference noise ratio of RIS ($SINR_{RIS}$) at $U$ using (1) is calculated as:

$$SINR_{RIS} = \frac{\sum_{i=1}^{N} |h_{s,i} h_{r,i}|^2 (rd)^{-\beta}}{\sum_{k=1}^{M} |h_{c,k}| l_k^{-\beta} \delta_k + \frac{1}{\rho}}, \tag{2}$$

where $\rho = \frac{P}{\sigma^2}$ is the SNR. By using the Shannon capacity formula, the ergodic capacity by an arbitrary user positioned at $U$, $C_{RIS}$ can be estimated as:

$$C_{RIS} = E\left[ log_2 \left( 1 + \frac{\sum_{i=1}^{N} |h_{s,i} h_{r,i}|^2 (rd)^{-\beta}}{\sum_{k=1}^{M} |h_{c,k}| l_k^{-\beta} \delta_k + \frac{1}{\rho}} \right) \right]. \tag{3}$$

Equations (2) and (3) consist of a variety of RV, e.g., $|h_{s,i}|^2$, $|h_{c,i,k}|^2$, $d$, $\delta_k$ etc.; therefore the closed-form expression of their probability density functions (PDFs) for $SINR_{RIS}$ is generally difficult to obtain, particularly for large values of N and M. A direct method to compute the average $SINR_{RIS}$ in this case would require at least $\mathcal{L}(N + M)$-fold integrations, where $\mathcal{L}$ is the total number of RV. Instead of performing time-consuming integration with respect to a large number RV in (3), a non-direct method using the following lemma is used to reduce the complexity [6]:

$$ln\left( 1 + \frac{u}{v + a} \right) = \int_0^\infty \frac{1}{z}\left( 1 - e^{-zu} \right) e^{-zv} e^{-za} \, dz. \tag{4}$$

Using (4), the expression in (3) can be re-expressed as

$$C_{RIS} = log_2 e \int_0^\infty \frac{1}{z} \left( 1 - \prod_{i=1}^{N} E\left[ e^{-z|h_{s,i} h_{r,i}|^2 (rd)^{-\beta}} \right] \right) \times \prod_{k=1}^{M} E\left[ e^{-z|h_{c,k}| l_k^{-\beta} \delta_k} \right] e^{-\frac{z}{\rho}} \, dz. \tag{5}$$

Assuming an independent and identically distributed (i.i.d.) of RVs, for a fully loaded traffic activity, (5) is reduces into:

$$C_{RIS} = log_2 e \int_0^\infty \frac{1}{z} \left( 1 - \prod_{i=1}^{N} M_i(z) \right) \times \prod_{k=1}^{M} M_k(z) e^{-\frac{z}{\rho}} \, dz, \tag{6}$$

Nakagami-m can be represented by the gamma distribution with m as the Nakagami fading index for useful signals from the base station and $m_i$ as the Nakagami fading index from co-channel interferers, where both m and $m_i \gg 1/2$. The Nakagami-m statistical model can be used to represent many fading conditions, such as the one-sided Gaussian distribution ($m = \frac{1}{2}$), and Rayleigh distribution when $m = 1$ as a special case. Rician distribution can also be closely approximated using the Nakagami-m when $m > 1$. As $m \to \infty$, the Nakagami fading channels converge to an AWGN channel. To simplify, assume $m_i = m$, then using the MGF of gamma distribution for the useful signals,

$$M_i(z) = \int_0^\infty \left( \frac{1}{1 + \frac{z}{m} x (rd)^{-\beta}} \right)^m \frac{m^m x^{m-1}}{\Gamma(m)} e^{-mx} \, dx = \left( \frac{m^2}{z(rd)^{-\beta}} \right)^m U\left( m, 1, \frac{m^2}{z(rd)^{-\beta}} \right), \tag{7}$$

where $\Gamma(.)$ Is the gamma function and $U(.)$ is the Triconomi confluent hypergeometric function. In the special case wherein Nakagami $m = 1$ (i.e., Rayleigh fading), with the help of ([12], (3.310)) and ([12], (3.353.5)), (7) becomes

$$M_i(z) = \frac{1}{z(rd)^{-\beta}} e^{\frac{1}{z(rd)^{-\beta}}} \Gamma\left( 0, \frac{1}{z(rd)^{-\beta}} \right), \tag{8}$$

where $\Gamma(x, y)$ is in complete gamma function. For $x = 0$, the incomplete gamma function is closely related to the exponential integral $\text{Ei}(z)$ ([12], (8.211/1)) as follows:

$$\Gamma(0, y) = -\text{Ei}(-y) + \frac{1}{2}\left[ln(-y) - ln\left(-\frac{1}{y}\right)\right] - ln(y). \tag{9}$$

Therefore, the MGF of $M_i(z)$ can be simplified as

$$M_i(z) = \frac{-1}{z(rd)^{-\beta}} e^{\frac{1}{z(rd)^{-\beta}}} \left[\text{Ei}\left(\frac{-1}{z(rd)^{-\beta}}\right)\right.$$
$$\left. -\frac{1}{2}\left[ln\left(\frac{-1}{z(rd)^{-\beta}}\right) - ln\left(-z(rd)^{-\beta}\right)\right] + ln\left(\frac{1}{z(rd)^{-\beta}}\right)\right]. \tag{10}$$

Assuming independent identically distributed (i.i.d.) links with the same stochastic characteristics, since it is a summation of the reflected signal of N elements of RIS (equal expected SNR, Nakagami m factor of all channel and distance), the total moment is obtained as:

$$\prod_{i=1}^{N} M_i(z) = M_i(z)^N. \tag{11}$$

For $M_k(z)$, since the $|h_{c,k}|^2$ is a gamma distribution gain, the total moment obtained is:

$$M_k(z) = \left(\frac{1}{1 + \frac{z}{m} l_k^{-\beta}}\right)^m. \tag{12}$$

For a random position of user, the PDF of the reference user $U$ schemes at radius d relative to RIS$_0$ is $p_d(d) = \frac{2d}{r^2} = \frac{2d}{\left(\frac{R}{3}\right)^2}$ for $0 \leq d \leq \frac{R}{3}$. Thus, new mathematical analysis for the ergodic capacity of the RIS model is as follows:

$$C_{RIS} = log_2 e \int_0^\infty \int_0^{\frac{R}{3}} \frac{1}{z}\left(1 - \prod_{i=1}^{N} M_i(z)\right) \times \prod_{k=1}^{M} M_k(z) \, e^{-\frac{z}{\rho}} \, \frac{2d}{\left(\frac{R}{3}\right)^2} dd \, dz. \tag{13}$$

where $\prod\limits_{i=1}^{N} M_i(z)$ and $M_k(z)$ are equal to (11) and (12), respectively. Equation (13) simplifies the evaluation of ergodic capacity by finding the MGF of $M_i(z)$ and $M_k(z)$ without having to deal with many RV. By fixing the value R, (13) simplifies the evaluation of the ergodic capacity of RIS in a single integral only. With this regard, (13) can be expressed in terms of weight and abscissa of a Laguerre orthogonal polynomial ([13], (25.4.45)), provides an efficient numerical evaluation method for the required average.

The ergodic capacity in (13) computed is under the assumption that all interferences coming from all other BS channels are active; thus, the number of interferences is constant. To investigate the effect of partially loaded BSs traffic conditions on the interfering cells, $\delta_k$ is a RV of on–off traffic with the expectation of MGF over the Bernoulli distribution, where we define $q$ as the probability that the channel is active, and $1 - q$ as the probability that the channel is not active. We include the MGF of the Bernoulli distribution, where the probability of the traffic activity $P_r(\delta_k = 1) = q$ and $P_r(\delta_k = 0) = 1 - q$,

From (13), the ergodic capacity is obtained as:

$$C_{RIS} = log_2 e \int_0^\infty \int_0^{\frac{R}{3}} \frac{1}{z}\left(1 - \prod_{i=1}^{N} M_i(z)\right) \times \prod_{k=1}^{M}(1 - q + q \cdot M_k(z)) \, e^{-\frac{z}{\rho}} \, \frac{2d}{\left(\frac{R}{3}\right)^2} dddz. \tag{14}$$

In order to calculate EE, the BS transmit power, $P_T$, including the hardware static power consumed to operate the BS circuit, $P_{BS}$ and RIS circuit power, $P_{RIS}$, make up the overall power consumed to operate the analyzed RIS-based system. It is worth noting that the RIS does not absorb any transmit power because its reflectors are passive-element devices that do not amplify or reducing the magnitude of the incoming signal directly.

Any amplification gain supplied by the RIS is produced by adjusting the phase shifts of the reflecting elements in such a way that the reflected signals are recombined with phase coherence. Different bit-resolution phase shifting patterns of each RIS element have different power consumption levels. The 3-, 4-, 5-, and 6-bit resolution consumed 1.5, 4.5, 6.0, and 7.8 mW of power, respectively [7]. The power dissipated on the whole surface of RIS with N identical reflecting components can be represented as $P_{RIS} = NP_n(b)$, where $P_n(b)$ specifies the power consumption of each phase shifter with $b$-bit resolution. The EE of the considered RIS-based system can then be expressed as [14]:

$$EE_{RIS} = \frac{E[C_{RIS}]}{P_T + P_{BS} + P_{RIS}}, \tag{15}$$

where the nominator in (15) is (14).

In order to measure the ASE, we introduce the fractional frequency reuse (FFR) scheme into the system model where a unity frequency reuse factor (FRF = 1) is assigned at the cell center while the cell edges are assigned FRF of three (FRF = 3). Let the total available bandwidth be partitioned into two disjoint sets of non-overlapping frequency bands. At the cell edges, the sub-band frequency is divided among them in such that the frequency allocated is different from their adjacent cells. We assume that all of the RISs at the cell edge serve the mobile user $U$ in the edge region using the maximized strategy as analyzed in the previous section. Then, the average area spectral efficiency can be defined as the maximum average data rates per unit bandwidth per unit area. Based on (14), we arrived at the following expression for the area spectral efficiency (ASE) for RIS at the cell edge in bits/s/Hz/km$^2$ as [15]:

$$ASE_{RIS} = q\frac{E[C_{RIS}]}{\pi R^2 (FRF)}. \tag{16}$$

### 2.2. Distributed Antenna System (DAS) System Model

DAS is an architecture wherein many antenna elements are geographically distributed and connect to a central BS in order to shorten the distance to the end users [14]. DAS provide coverage to the nearby users, and they are connected to the central BS via high-speed low-latency links, fiber or microwave [16,17], or connected to the CPU at the central BS through fronthaul links [18]. Intuitively, uniformly distributed antenna clusters in each cell provide a significantly higher chance for a cell-edge user to be close to some BS antenna in the neighboring cell [19]. Here, we also assumed that all users have an identical number of antennas serving them [20].

Consider the same system model as the RIS model, but instead of user $U$ receiving signals from the nearest DAS, user $U$ is receiving signals from all DASs collocated with the BS [1] as in Figure 2. Hence, the signal received by the reference user $U$ in the presence of C collocated DAS and CCI from $M$ neighboring cells can be expressed as:

$$y_{DAS} = \sqrt{P}x_0\left(\sum_{i=1}^{C} h_i \sqrt{r_i^{-\beta}}\right) + \sum_{k=1}^{M} \sqrt{P}x_k\left(\sum_{i=1}^{C} h_{i,k}\sqrt{l_{k,i}^{-\beta}}\right)\delta_k + w_0, \tag{17}$$

where the first term is the useful signal while the second term is the interfering signal, and $w_0$ represents the additive white gaussian noise (AWGN) that satisfies $E\left[|w_0|^2\right] = \sigma^2$. $x_0$ and $x_k$ are the transmitted signals from useful and interfering BS, respectively, while $P$ is the transmitted power from BS. $|h_i|^2$ and $|h_{i,k}|^2$ are the channel gains from home cell DASs to $U$ and first-tier CC DASs to $U$, respectively. $\beta > 2$ is the path loss exponent and $\delta_k$ is the traffic-activity factor. The instantaneous SINR at $U$ for DAS system is defined as:

$$SINR_{DAS} = \frac{\sum_{i=1}^{C} |h_i|^2 r_i^{-\beta}}{\sum_{k=1}^{M} \sum_{i=1}^{C} |h_{i,k}|^2 l_{k,i}^{-\beta} \delta_k + \frac{1}{\rho}}. \tag{18}$$

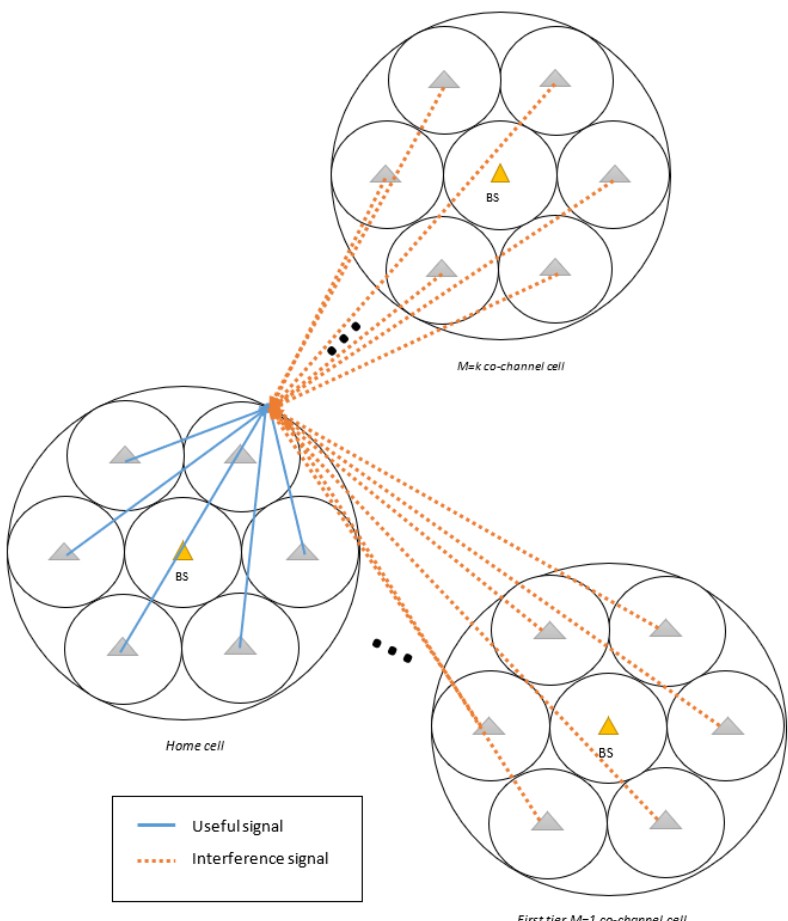

**Figure 2.** DAS-aided system model.

From the Shannon capacity formula, the ergodic capacity achieved by user $U$ can be estimated as:

$$C_{DAS} = E\left[log_2\left(1 + \frac{\sum_{i=1}^{C}|h_i|^2 \, r_i^{-\beta}}{\sum_{k=1}^{M}\sum_{i=1}^{C}|h_{i,k}|^2 l_{k,i}^{-\beta}\delta_k + \frac{1}{\rho}}\right)\right]. \quad (19)$$

A non-direct method based on (4) is used to simplify the expression of the ergodic capacity into MGF terms. Including the PDF of reference of $U$ schemes at radius $d$ and angle $\theta$ relative to DAS given by (9) and $p_d(d) = \frac{2d}{\left(\frac{R}{3}\right)^2}$ for $0 \leq d \leq \frac{R}{3}$, the ergodic capacity can be simplified as:

$$C_{DAS} = log_2 e \int_0^\infty \int_0^{\frac{R}{3}} \int_0^{2\pi} \frac{1}{z}\left(1 - \prod_{i=1}^{C} M_2(z)\right) \times \prod_{k=1}^{M}\prod_{i=1}^{C} M_3(z) \, e^{\frac{-z}{\rho}} \, \frac{2d}{\left(\frac{R}{3}\right)^2}\delta_k \, \frac{1}{2\pi}d\theta d d dz, \quad (20)$$

where

$$M_2(z) = \left[1 + \frac{z}{m}\left((x_i - dcos(\theta))^2 + (y_i - dsin(\theta))^2\right)^{\frac{-\beta}{2}}\right]^{-m}, \quad (21)$$

and

$$M_3(z) = \left[1 + \frac{z}{m}\left((x_{i,k} - dcos(\theta))^2 + (y_{i,k} - dsin(\theta))^2\right)^{\frac{-\beta}{2}}\right]^{-m}. \quad (22)$$

where $x$ and $y$ are the coordinate of DAS in a Cartesian plane.

Regarding the effect of partially loaded traffic or traffic activity on the interfering cells, $\delta_k = 1$ if channel $k$ is active, $\delta_k = 0$ otherwise. We include MGF of the Bernoulli distribution,

where the probability of the traffic activity $P_r(\delta_k = 1) = q$ and $P_r(\delta_k = 0) = 1 - q$ in the average capacity, obtained as:

$$C_{DAS} = log_2 e \int_0^\infty \int_0^{\frac{R}{3}} \int_0^{2\pi} \frac{1}{z} \left(1 - \prod_{i=1}^C M_2(z)\right) \times \prod_{k=1}^M \prod_{i=1}^C [1 - q + q.M_3(z)] \, e^{\frac{-z}{\bar{\rho}}} \, \frac{2d}{\left(\frac{R}{3}\right)^2} \, \frac{1}{2\pi} d\theta d d z. \quad (23)$$

Each cell in the proposed system has one BS in the center, which is connected to the DASs through front-haul fiber optic cables. The BS power usage is considered in addition to the DAS's. A DAS system power consumption can be divided into two categories: total hardware static power consumption and power consumed by the power amplifier (PA) [8], which can be expressed as:

$$P_{Total} = (1 + \tau) \sum_{i=1}^C P_{T,i} + P_{BS} + \varepsilon E[C_{DAS}], \quad (24)$$

where $\tau = \frac{\varsigma}{\wp} - 1$ with $\varsigma$ is the peak-to-average ratio (PAR) and $\wp$ is the PA drain efficiency, while $\varepsilon$ represents the dynamic power consumption per throughput. It is assumed that, if the transmit power for each DAS is equal, then $\sum_{i=1}^C P_{T,i} = CP_T$. Therefore, the EE of the proposed DAS system can be calculated as [15]:

$$EE_{DAS} = \frac{E[C_{DAS}]}{P_{Total}}. \quad (25)$$

From (23), using the same scheme of FFR as in RIS system model, the expression for ASE in bits/s/Hz/km² for DAS at the cell edge is as follows [16]:

$$ASE_{DAS} = q \frac{E[C_{DAS}]}{\pi R^2 (FRF)}. \quad (26)$$

## 3. Numerical and Simulation Results

In this section, the numerical and simulation results are presented. The simulation parameters used are similar to [21], for the values for the static BS power, power amplifier efficiency and dynamic power consumption per unit throughput; other parameters are shown in Table 1. As per Figures 1 and 2 earlier, we assume the cellular system radius R = 3 km and we vary the SNR from −20 dB to 20 dB. The numerical results obtained by using the theoretical approach using Equation (13) are verified by $10^5$ Monte Carlo iterations based on Equation (3) for RIS.

**Table 1.** Simulation parameter values.

| Parameter | Value |
|---|---|
| Radius, R | 3 km |
| Number of co-channel cell, M | 6 |
| Number of RIS/DAS per cell, C | 6 |
| Transmit power, $P_T$ | 1 W |
| Static BS power, $P_{BS}$ | 7.9843 W |
| Static RIS element power, $P_n(b)$ | 6.0 mW |
| Number of RIS element, N | 10 until 100 |
| Power amplifier efficiency, $\tau$ | 38% |
| Dynamic power consumption per unit throughput, $\varepsilon$ | 0.1 W/bps |
| Nakagami index, m | 1 |
| Path loss exponent, β | 2.5 |
| Activity factor, $q$ | 0.1 to 1 |
| Cell edge FRF | 3 |
| SNR | −20 dB to 20 dB |

From Figure 3, when the values of the interference number and power are fixed, the antenna for RIS is set with N = 10, 30, 50 and 100, while for DAS, we set omni-channel



antenna per DAS cell. We can see that the ergodic capacity performance improves as the number of elements in RIS increase. The RIS-aided system also provides a higher capacity below 10 dB SNR; for instance, at 0 dB, the RIS ergodic capacity for N = 100 is at 4.8 bits/s while for DAS, it is at 2 bits/s. At a high interference area and high SNR region, 20 dB, the RIS average capacity peak is 5.5 bits/s while that of the DAS-aided system is 5 bits/s, with smaller difference in comparison to 0 dB. This is because users in the DAS system suffered from various combinations of interfering signals from co-channel DASs. In contrast, the RIS system is assumed to be capable of reflecting away the undesired signals within its own cellular system, and interferers only come from other co-channel BSs. However, for DAS, interferers can come from other co-channel interferers from other cells, either from DASs or other BSs.

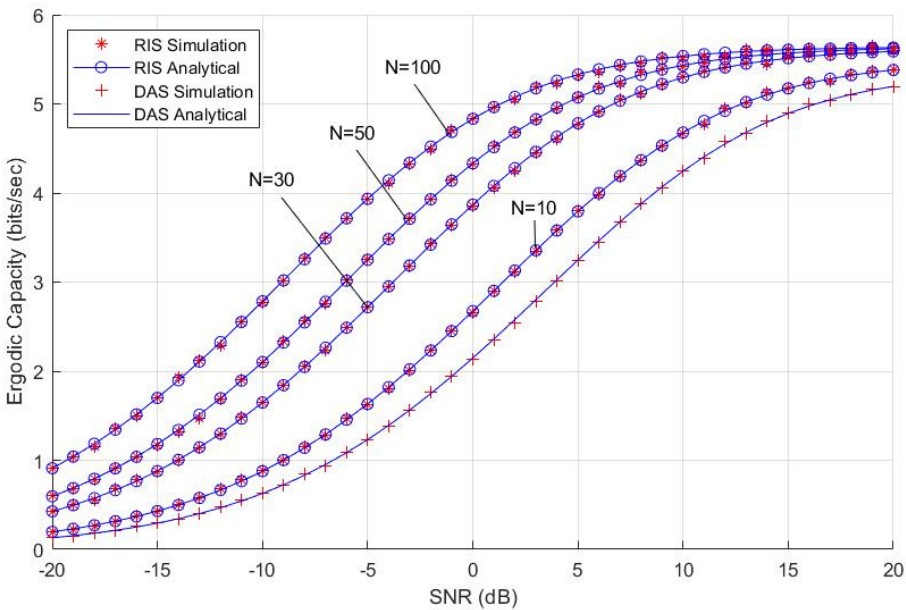

**Figure 3.** Ergodic capacity vs. SNR with varied number of RIS element, N.

Figure 4 reveals that the greater the attenuation to the useful signals and interference signals (i.e., having a higher path-loss exponent, β), the greater the ergodic capacity in comparison to the signals with a lower β. No difference in ergodic capacity achieved by the user from −20 dB until 0 dB SNR for the RIS system at different values of path loss exponent, β, while the ergodic capacity increases almost linearly with respect to the SNR. However, as the value of SNR increases past 0 dB, it is obvious that the performance is much better in the higher value of path-loss exponent due to the interfering signal strength received by the user becoming weaker. For a very large distance of co-channel sources (12 km in this case), the interfering signal becomes less significant as the value of path loss exponent increases. Hence, the received signal improves for higher values of path loss exponent due to less interference. In high-SNR areas at 20 dB, the average capacity tends to saturate between 7 and 10 bits/s for RIS, while for DAS, the average capacity to saturate is between 7 and 8 bits/s.

Figure 5 shows the ergodic capacity against SNR curves with different traffic activity factor, $q$. The six downlink interferers, for a zero-loaded system or traffic activity factor ($q = 0$) of interfering cells, have the highest average capacity. It is commonly known that a signal has a maximal performance when there is zero presence of interfering signals. Comparing the RIS system and DAS, as the traffic activity factor $q$ increases, the average capacity of the RIS system drops drastically compared to the DAS. This is because for each interfering signal from six co-channel BSs in the RIS system, the combined interference signal's power is much higher than the useful signals, thereby reducing SINR as in Equation (3) in the high SNR region and as $q$ tends to approach 1 (fully loaded system).

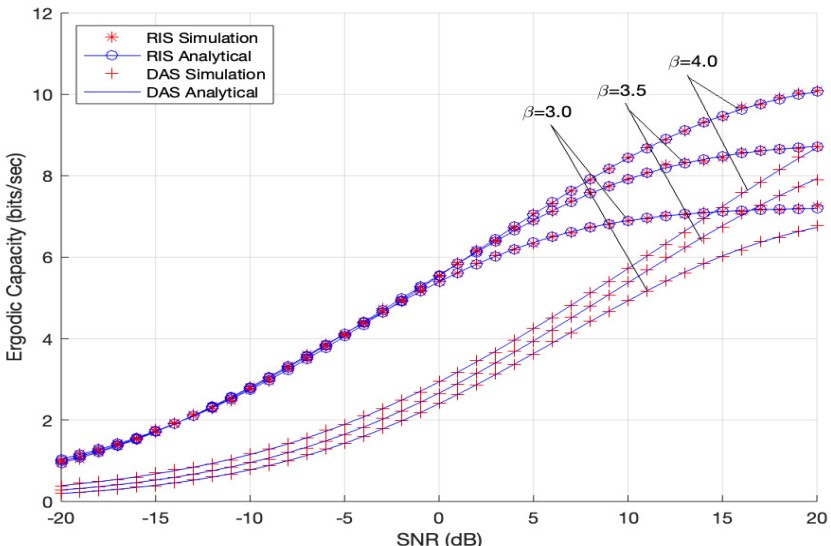

**Figure 4.** Ergodic capacity vs. SNR with varied path loss exponent, β.

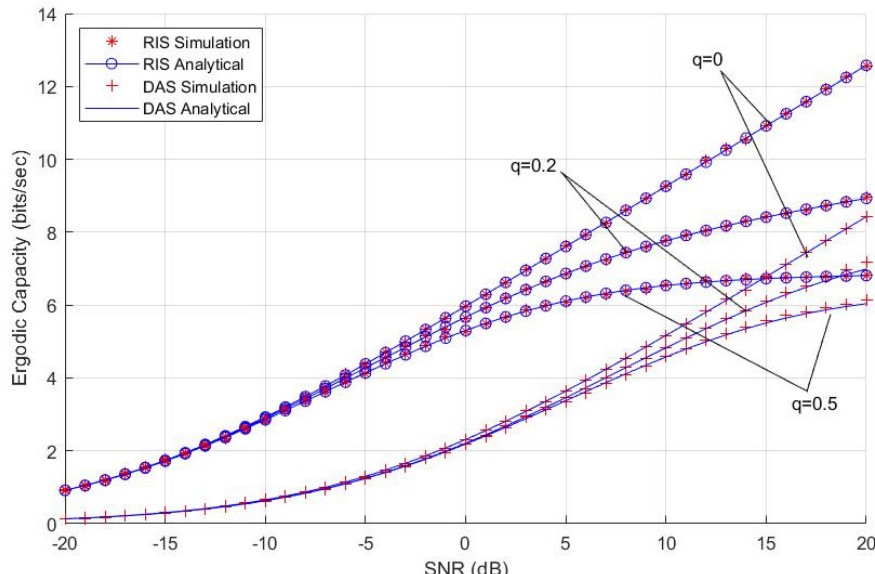

**Figure 5.** Ergodic capacity vs. SNR with varied traffic activity factor, $q$.

Figure 6 shows that as the number of RIS elements N increases from N = 10 to N = 100, the EE performance for RIS also increases, accordingly. This is because the power consumption from RIS elements is very low due to the passive use of power for the RIS metasurface. As per EE in Equation (15), the denominator is largely dominated by the base station power rather than the RIS passive elements power. Thus, the effect of RIS hardware power consumption is still relatively low, even when the number of RIS elements increases to N = 100.

In Figure 7, for FRF = 3 for the cell edge, we can see that the partially-loaded system has lower ASE than a fully-loaded system, for both RIS and DAS, since the ASE increases as the active channels in the systems increase. From (16) and (25), although the individual channel achieved higher rates in a partially loaded system due to the lower CCI; this effect is offset by the fact that there are fewer active channels $q$ in the system. By comparing the two systems, we can see that the RIS system provides better ASE or data rates supported per cell-edge in bits/s/Hz/km$^2$ than the DAS system for different values of active channels in a cell, $q$. The results show that the channel efficiency per unit area for RIS is better than DAS, mainly due to the better average capacity of RIS caused by the summation factor of

the number of antennas N in nominator in (2), and less CCI from other RISs located in $k_{th}$ cells (in comparison to DAS). Figure 7 also shows that, in a high SNR region, the gap in ASE performance between both systems narrows by about 0.005 bits/s/Hz/km$^2$ while at 0 dB or less, the gap in ASE between the two systems widens to 0.03 bits/s/Hz/km$^2$. From design perspective, both systems have the highest ASE when all available channels for the cell-edge is being utilized.

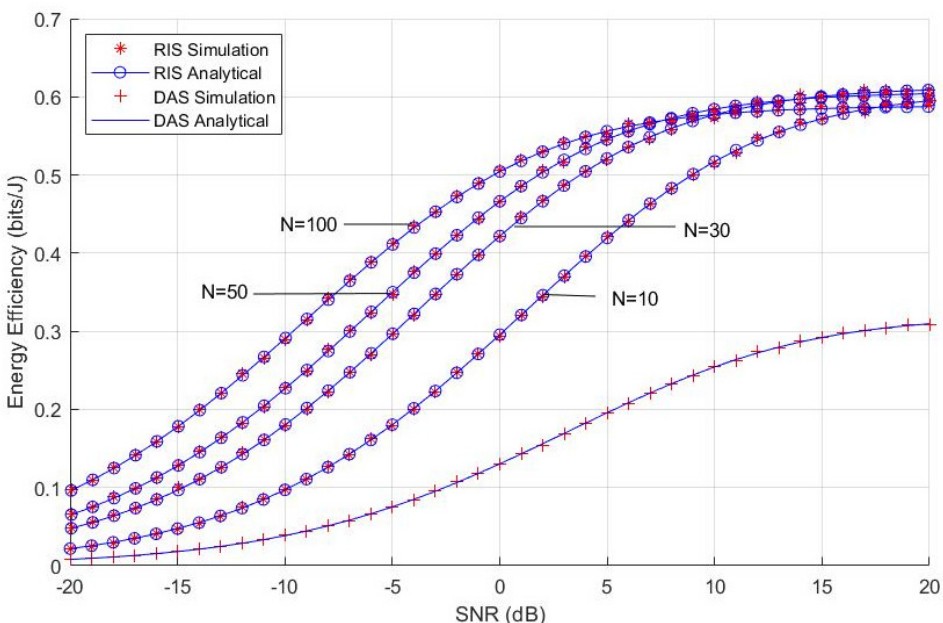

**Figure 6.** Energy efficiency vs. SNR with varied N.

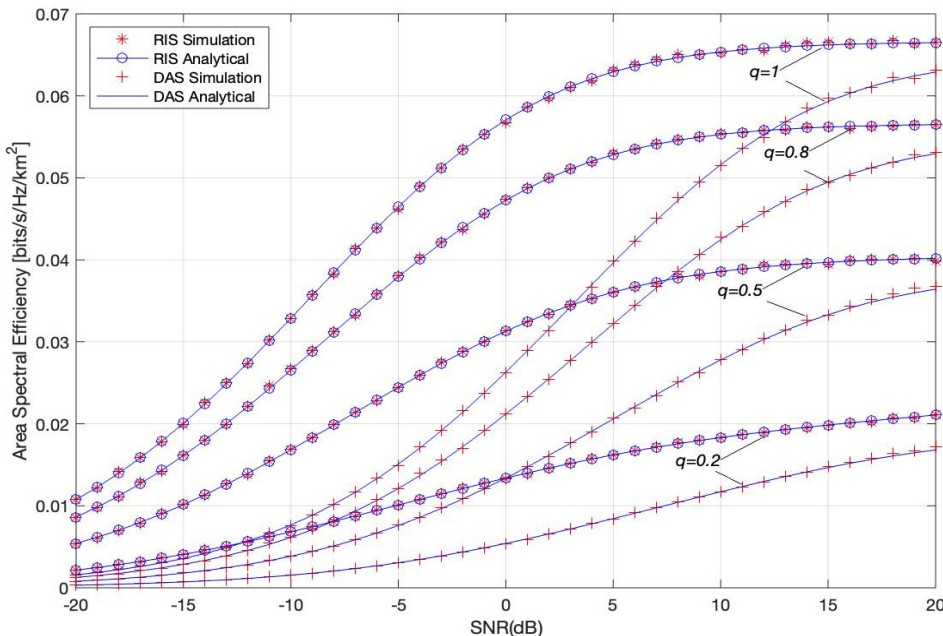

**Figure 7.** Area spectral efficiency vs. SNR with variable *q*.

Based on Figures 3–7, it is evident that our analytical method using MGF work almost perfectly and was validated using $10^5$ Monte Carlo simulations.

## 4. Conclusions

This study investigated a correlation of execution in the RIS-aided system cell framework and DAS-aided system cell framework, wherein a simple and precise structure for

the performance measurement under realistic channel presumptions was introduced. The analysis of the downlink ergodic capacity with regards to the RIS framework and DAS framework under the Nakagami-m fading channel and path loss attenuation, with the presence of interference from co-channel BSs, was simplified by utilizing an MGF-based approach. It is observed that, under ideal RIS channel assumptions and conditions, utilizing RIS at the cell edge provides better ASE than DAS, especially when the traffic activity factor (co-channel interferences) decreases. The EE for RIS improves as the number of antennas, N, increases, in comparison to DAS. However, at a high SNR of 20 dB, the ASE gap for both systems decrease to 0.005 bits/s/Hz/km$^2$, while at a lower SNR, the difference in ASE between the two systems is greater. Both systems have highest ASE when the channels in the system are all in used.

**Author Contributions:** N.A.K., A.M. and M.Y.B.A. were involved in the analysis and system model design; N.A.K. and A.M. produced and verified the analysis and simulation; N.A.K., A.M. and M.Y.B.A. carried out the formal analysis; N.A.K. and A.M., writing—original draft preparation; M.Y.B.A., A.A.A. and S.Y., writing—review and editing; M.Y.B.A., help in securing the funding. All authors have read and agreed to the published version of the manuscript.

**Funding:** This research was funded by Telekom Malaysia Research and Development (TMRND), grant number MMUE/220001; and Ministry of Higher Education, Malaysia, under the Fundamental Research Grant Scheme (FRGS) number MMUE/190229 (FRGS/1/2020/TK02/MMU/03/1).

**Institutional Review Board Statement:** Not applicable.

**Informed Consent Statement:** Not applicable.

**Data Availability Statement:** Not applicable.

**Conflicts of Interest:** The authors declare no conflict of interest.

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
