# Peer review of "Performance Evaluation of Reconfigurable Intelligent Surface against Distributed Antenna System at the Cell Edge"

_electronics, doi:10.3390/electronics11152376_

Round 1

Reviewer 1 Report

1. The paper structure should be described at the end of the introduction section.

2. The introduction should indicate the research gaps and research goals. What is the main question addressed by the research?

3. What does it add to the subject area compared with other published material?

4. The authors are suggested to cite related works to illustrate your equations. And should make a connection with these equations. Why do you use related theorem? The readers do not know why you use these equations? It is hard to realize these equations are related to your works.

5. The resolution is too poor in Fig. 1

6. What specific improvements could the authors consider regarding the methodology?

7. In the Methodology section, the authors should use a flowchart to describe the proceeding flowchart. Please use a standard flowchart to illustrate the process. For example, use an elliptical graph to illustrate the “start” and “end”, use the diamond graph to illustrate the judgment events.

Reviewer 2 Report

Authors did a hasty draft without looking into the depth of the contents.

a) Lot of abbreviations are used without any extension.

b) So many terms are used which make the paper a little bit baffling.

c) There is no explanation why authors choose those simulation parameters.

d) Abstract and conclusion are same which is not acceptable.

e) The work needs full organization.

f) The Figures are explained very superficially without laying emphasis on the underlying principle.

g) Some of the parameters used in the equations are not described.

h) Author should get their paper checked by a native English speaker or editing services as most of the contents are not understood at all. Many of them are verbose.

Round 2

Reviewer 1 Report

The authors have fixed the previous concerns.

Author Response

Thank you very much for your kind review

Reviewer 2 Report

There are still missing points.

a) There are terms still not properly specified.

b) The conclusion and abstract remain elusive.

c) Results are not matching with provided figure.

d) There are so many redundant portions newly added. There is no Section VI in the revised MS although it is mentioned in the MS.

e) Explanations are not still up to the mark. 

Round 3

Reviewer 2 Report

Some minor syntax errors are there. Sporadic use of abbreviations be avoided.

Author Response

Thank you for your kind review. We have corrected the syntax errors.